# Long-Term Care and Follow-Up in Laryngeal Cancer Patients: A Multicenter Retrospective Analysis

**DOI:** 10.3390/jpm13060927

**Published:** 2023-05-31

**Authors:** Blažen Marijić, Filip Tudor, Stefan Janik, Stefan Grasl, Florian Frommlet, Diana Maržić, Ita Hadžisejdić, Jelena Vukelić, Tamara Braut, Marko Velepič, Boban M. Erovic

**Affiliations:** 1Institute of Head and Neck Diseases, Evangelical Hospital, 1180 Vienna, Austria; blazen.marijic@uniri.hr; 2Department of Otorhinolaryngology, Head and Neck Surgery, Clinical Hospital Center Rijeka, 51000 Rijeka, Croatia; filip.tudor@medri.uniri.hr (F.T.); tamara.braut@uniri.hr (T.B.); marko.velepic@medri.uniri.hr (M.V.); 3Faculty of Medicine, University of Rijeka, 51000 Rijeka, Croatia; diana.marzic@uniri.hr (D.M.); ita.hadzisejdic@uniri.hr (I.H.); jelena.vukelic@uniri.hr (J.V.); 4Department of Otorhinolaryngology, Head and Neck Surgery, Medical University Vienna, 1090 Vienna, Austria; stefan.janik@meduniwien.ac.at (S.J.); stefan.grasl@meduniwien.ac.at (S.G.); 5Center for Medical Statistics, Informatics and Intelligent Systems, Section for Medical Statistics, Medical University Vienna, 1090 Vienna, Austria; florian.frommlet@meduniwien.ac.at; 6Department of Audiology and Phoniatrics, Clinical Hospital Centre Rijeka, 51000 Rijeka, Croatia; 7Clinical Department of Pathology and Cytology, Clinical Hospital Center Rijeka, 51000 Rijeka, Croatia

**Keywords:** long-term care and follow-up, laryngeal carcinoma, secondary primary tumor, recurrence, survival

## Abstract

Purpose: We conducted an outcome analysis on surgically treated laryngeal squamous cell carcinoma (LSCC) patients. Methods: A multicenter retrospective study with 352 patients was analyzed. A new nomogram that incorporates age, T- and N-classification, and treatment was created. Results: Recurrence was observed in 65 (18.5%) patients after a mean time of 16.5 months. After 60 months, 91 (25.9%) of patients developed secondary primary tumors (SPT), most commonly in the lungs (*n* = 29; 8.2%) followed by other head and neck cancers (*n* = 21; 6.0%). Notably, the mean time to occurrence of secondary head and neck cancers was twice that of lung cancer (101.1 vs. 47.5 months). Conclusion: Recurrent disease is less common in LSCC patients and appears much earlier than SPT. Because one in every four laryngeal cancer patients develops SPTs within 5–10 years, long-term care and follow-up, including imaging studies, are highly recommended. The nomogram was useful for estimating survival.

## 1. Introduction

Despite continuous advances in medicine over the past decades, laryngeal cancer remains one of the most challenging malignancies of the upper respiratory tract with unsatisfactory outcomes [1,2]. Death is most frequently linked to recurrent disease and, when excluding cardiovascular causes, secondary primary tumors (SPTs) represent the main determinants of overall survival [3]. According to NCCN guidelines (version 1.2023), the majority of tumor recurrences and SPTs occur in the first two postoperative years [4]. Although SPTs can occur anywhere in the human body, previous studies reported that SPTs localized in the lung are the main drivers of death during follow-up of patients with laryngeal malignancies [5,6,7]. Therefore, great efforts have been undertaken to improve necessary controls and regular checkup appointments in laryngeal squamous cell carcinoma (LSCC) patients. Although it is still unknown whether earlier detection of SPTs leads to increased survival, detecting the tumor in an earlier stage will undoubtedly make treatment easier [4]. To date, however, there is no consensus or set of guidelines to facilitate the early detection of SPT. 

Herein, we report our experiences regarding surgical treatment of laryngeal malignancies with a particular interest in the occurrence of secondary malignancies. It was, therefore, the purpose of the study to evaluate patterns and causes of treatment failures, the occurrence of recurrences, and secondary malignancies as well. Based on these data, it was the secondary goal to identify predictors of poorer outcomes and to incorporate these into a nomogram for predicting survival.

## 2. Materials and Methods

### 2.1. Study Cohort

We performed a multicentric retrospective study including 352 patients who underwent surgical treatment for LSCCs (Figure 1). All patients were treated between January 1999 and October 2020 at (i) the Department of Otorhinolaryngology, Head and Neck Surgery of the Medical University of Vienna, Austria, (ii) the Institute of Head and Neck Diseases, Evangelical Hospital, Vienna, Austria, and (iii) at the Department of Otorhinolaryngology, Head and Neck Surgery of the Clinical Hospital Center Rijeka, Croatia. The study was conducted according to the guidelines of the Declaration of Helsinki and approved by the Institutional Review Boards (EK No. 1758/2017 and No. 003-05/17-120). 

Clinical data were retrospectively obtained from electronic patient records. Data were collected regarding basic patients’ characteristics (sex, age, and nicotine and alcohol use), primary tumor details (TNM classification and stage according to the 8th edition of the American Joint Committee on Cancer (AJCC)/International Union Against Cancer (UICC) staging system [8], and grading), treatment information (extent of surgery, resection margins, adjuvant therapy), oncological outcome (recurrence, survival), secondary primary tumor details (time to the occurrence, type, and site of SPTs) and follow-up. The interval between visits during this follow-up was in alignment with the recommendations of the National Comprehensive Cancer Network (NCCN) [4].

### 2.2. Recurrent Disease Versus Secondary Primary Tumor

To distinguish between recurrent disease and SPT, we used the following criteria: (i) histology different from squamous cell carcinoma (SCC), (ii) more than 3 years between the occurrence of the second malignancy and primary tumor treatment, and (iii) ≥2 cm of normal epithelium between the primary and secondary tumor if the secondary tumor occurred near the primary tumor [9,10,11].

Since secondary primary lung cancers frequently occur in laryngeal cancer patients, it was necessary to further differentiate between pulmonary metastases and primary lung cancer. Multiple lung nodules were principally rated as pulmonary metastases, while a solid lesion was more likely for primary lung cancer. Moreover, a nodule localized in the middle or distal part of the lung also suggests lung metastasis, while hilar or endobronchial lesions are more likely to indicate a secondary primary tumor. The criteria of Rott et al., which state that endobronchial metastases are quite uncommon and were only detected in 0.68% of cases [12], were adopted in our research. The same study described that extrabronchial metastases more frequently affect the surface epithelium of the lungs [12]. 

If radiological characteristics were inconclusive, histological biopsy and verification were performed. In addition, since pulmonary lesion/lesions in early-stage laryngeal cancers with the absence of regional lymph nodes are common, secondary primary lung cancer is more likely than pulmonary metastasis [12,13,14], since skip metastases are rare.

### 2.3. Statistical Methods

Statistical analyses were performed using SPSS version 27.0 software (IBM SPSS Inc., Armonk, NY, USA). Unless otherwise specified, data are reported as mean ± standard deviation (SD). Descriptive statistics were used for the analysis of demographic and clinical data. The Chi-square test was used to investigate the association between nominal variables. An unpaired Student’s *t*-test was used to compare the means of two independent groups with normal (Gaussian) distributions. Kaplan–Meier analyses and log-rank tests were assessed for univariate outcome analysis. Overall survival was defined as the period reaching from the start of therapy to the end of follow-up or death of any cause. Death of any cause represents an event, while unknown outcomes or losses in follow-up were counted as censored events. Cancer-specific survival (CSS) was defined as the time between the start of treatment and death from cancer. Unrelated deaths, deaths for unknown reasons, or deaths caused by another malignant disease were censored events. The recurrence-free survival (RFS) was calculated only in patients who had undergone complete surgical resection (R0) from the date of surgery to the date of recurrence and had complete recurrence status information. Uni- and multivariate Cox regression analyses were used to evaluate the prognostic impact of different clinical variables on outcomes. Hazard ratios (HRs) and corresponding 95% confidence intervals (CIs) are indicated. All tests were performed two-sided and *p*-values below 0.05 were considered statistically significant. To create a nomogram, we performed variable selection among all potential predictor variables and some pairwise interactions using logistic regression models. Stepwise backward elimination based on the Akaike information criterion (AIC) was applied to obtain the final best multivariable logistic regression model, which was visualized with a nomogram using the R package “rms” [15].

## 3. Results 

### 3.1. Study Cohort

In total, 352 patients were analyzed, comprising 328 (93.2%) males and 24 (6.8%) females with a mean age of 62.5 ± 9.2 years. All patients were diagnosed with SCC. Here, 81 (23%) T1, 53 (15.1%) T2, 129 (36.6%) T3, and 89 (25.3%) T4a tumors were noted, with positive neck nodes in 71 (20.2%) patients (Table 1). Considering the extent of surgery, we noted 102 (29%) partial laryngectomies, 191 (54.3%) total laryngectomies, 49 (13.9%) partial laryngopharyngectomies, and 10 (2.8%) total laryngopharyngectomies. Furthermore, 308 (87.5%) patients underwent primary surgery, while 44 (12.5%) underwent salvage surgery. The cohort of patients who were irradiated postoperatively and those who were not were almost equal (*n* = 180; 51.5% versus *n* = 172; 48.9%). The total dose of radiation was 60–66 Gy. After the surgery, patients were presented to the multidisciplinary tumor board and, in a few cases, the decision was made for chemoradiotherapy as adjuvant therapy. Platin-based chemotherapy was used, which is the current standard of care [16]. Further socio-demographic data are shown in Table 1.

### 3.2. Occurrence of Secondary Primary Tumor

SPTs were diagnosed in 91 (25.9%) patients, with SPTs found in 26.6% (34/128) of stage I–II laryngeal cancer patients and 25.4% (57/224) of stage III–IV patients (*p* = 0.899). Therefore, the risk for SPT did not statistically differ between early and advanced stages of laryngeal cancer (Figure 1). SPT occurred mostly in the lungs (*n* = 29; 8.2%) followed by head and neck tumors of other subsites (*n* = 21; 6%; Table 2). The mean and median time of onset of the secondary tumor was available in 66 (72.5%) of 91 patients and was 60.48 and 54.4 months, respectively (range: 0.2–160.97 months). Furthermore, 57.6% (*n* = 38) and 89.4% (*n* = 59) of those experienced secondary carcinomas during the first 5 and 10 years after a laryngeal cancer diagnosis, respectively. As shown in Figure 2, we noticed a peak in the second year after laryngeal cancer treatment in which 11 patients had 3 lung tumors, while one had pancreatic, kidney, rectal, urinary bladder, esophageal, brain, and mesothelioma tumors, as well as non-Hodgkin lymphoma. Additionally, 4 of the 11 patients (36%) developed SPTs (lung and esophageal cancer) that are known to be associated with smoking and alcohol consumption. After the second year, the incidence curve flattens but remains constant until the tenth year of follow-up (Figure 2). Most notably, secondary head and neck cancers were noticed with a mean time of 101.1 months after initial diagnosis, which was two times longer compared to a diagnosis of secondary lung cancer (47.5 months) or carcinomas of other localizations (54.8 months). SPTs were treated based on the multidisciplinary team’s decision for each location and according to the protocol for each tumor. Appendix A shows the treatment modality for each SPT. Comparison of clinical features with secondary head and neck and lung tumors are summarized in Table 3. 

### 3.3. Tumor Recurrence

Recurrent disease was observed in 65 (18.5%) patients, of which 21 (6%) patients had local, 26 (7.4%) regional, and 18 (5.1%) distant recurrence (or distant metastases). The mean (median) time between laryngeal cancer and recurrence was 16.5 (12.6) months. Local, regional, and distant recurrences occurred in 15.5, 14, and 21.3 months, respectively. Recurrent disease was treated in accordance with the multidisciplinary team’s decision for head and neck cancers. Appendix A shows the treatment modality for each tumor recurrence. The significant impact of different clinical variables, such as T-classification or N-classification, on the type of recurrence is shown in Table 4.

### 3.4. Oncological Outcome

The mean follow-up time was 63.8 months. At the end of the observational period, 188 (53.4%) patients were alive and 164 (46.6%) were deceased. Causes of mortality were associated with recurrent disease in 62 (17.6%) cases, with SPTs in 45 (12.8%) and other causes (cardiovascular and long-term diseases) in 57 (16.2%) subjects. The predictive effect of different clinical variables was evaluated regarding overall survival (OS), cancer-specific survival (CSS), and recurrence-free survival (RFS), as shown in Table 5. 

In particular, age > 62.5 years (HR 1.61; *p* = 0.007), stage III–IV diseases (HR 3.19; *p* = 0.032), salvage surgeries (HR 2.23; *p* = 0.001), tumor recurrence (HR 5.05; *p* < 0.001), and SPTs (HR 1.44; *p* = 0.039) represented independent worse prognostic factors for OS. The impact of tumor recurrence and SPT on OS is illustrated in Figure 3A,B. However, positive neck nodes (HR 3.14; *p* < 0.001), salvage surgery (5.13; *p* < 0.001), and tumor recurrence (HR 33.3; *p* < 0.001) were independent poor prognosticators for CSS. Salvage surgery (HR 3.92; *p* < 0.001) and lymph node involvement (HR 2.23; *p* = 0.006) were also independent worse prognostic factors for RFS. 

### 3.5. Nomogram

Finally, we calculated a nomogram based on T-classification (T1, T2, T3, T4a), N-classification (N0 versus N1–3), age, and salvage situation (yes versus no) as significant factors for OS. Our nomogram indicates that elderly LSCC patients, with advanced T- and N-classification, who need to undergo salvage laryngeal surgery, show the worst OS. Applying our nomogram for a 60-year-old patient with a T3N1 laryngeal cancer who undergoes salvage surgery indicates a 1-, 3-, and 5-year OS of 60%, 30%, and 20%, respectively (Figure 4). 

## 4. Discussion

The first report on SPTs in head and neck cancer patients is more than 130 years old [17]. Despite advances in diagnostic, therapeutic, and follow-up practices, SPTs remain among the leading causes of mortality [18]. In general, SPTs occur in 7–26% of patients up to 25 years after the primary diagnosis of HNSCC [3,6,19,20]. In a meta-analysis published by Haughey and coworkers including 40,287 patients with HNSCC, the overall prevalence of SPT was 14.2% and tumors occurred mostly in a metachronous way [21]. Laryngeal carcinoma carries an excellent clinical outcome and prolonged survival if diagnosed in an early stage and treated appropriately [22].

However, prolonged overall survival also increases the risk for SPTs, which again may cause poor outcomes or death [18]. Advanced-stage laryngeal cancers and SPTs often require multimodal treatment as they tend to lead to tumor relapse and subsequently unfavorable OS [18,23]. SPTs occur more frequently in those parts of the human body that are exposed to the same etiological factors as laryngeal cancer. This can be partly explained by the concept of field cancerization, which has been reported a number of times before, with smoking, alcohol abuse, and irradiation cited as the main risk factors [24].

Our study showed an incidence of SPTs of almost 25%, which lies at the upper edge among published data [6,20]. Furthermore, almost a third of patients (32,1%) with advanced laryngeal cancer experienced SPT. As expected, the vast majority of SPTs occurred in the respiratory and upper digestive tract, with the most prevalent localization beng within the lungs, followed by the esophagus. The explanation of the high incidence of secondary pulmonary carcinomas may represent the “one airway, one disease theory”. Nonetheless, neither the rich connection of the lymphatic organs between the cervical and thoracic regions nor the venous drainage pathways leading from the neck to the lungs should be neglected [25,26]. This raises suspicion that some SPTs can be misinterpreted while they are actually distant metastases. For this reason, we included strict guidelines to distinguish recurrences from SPTs [9,10,11]. Nonetheless, we are aware of the fact that this is a limitation of the study and that only genetic profiling could provide for the precise differentiation between metastatic lesions and secondary tumors [27,28]. Some authors reported incidences of lung SPTs among overall SPTs of up to 39% [29,30] which is similar to our data (29/91 patients or 32%). Among various surveillance imaging tools, FDG PET/CT proved to be the most sensitive [31,32,33]. According to some authors, a 12-month PET scan revealed tumor recurrence or SPT in 10% of cases [34]. Interestingly, SPT was discovered in a distant site in most asymptomatic patients, with half of the tumors found in the lungs [35]. Given the fact that these secondary primary tumors reflect a poor clinical course with an unfavorable prognosis, imaging methods and regular pulmonary examinations are essential and must be regularly performed during follow-ups. 

Equally, the high incidence of metachronous esophageal malignancies can be explained by the proximity of anatomical location as well as by the field cancerization theory [24,36]. This strongly underlines the need for the inclusion of the esophagus into the pool of regions at higher risk that requires strict evaluation [25]. 

There are no consensus guidelines on the frequency and modality of regular post-treatment imaging in the asymptomatic patient, according to the NCCN [4]. All proposed guidelines are more or less aimed at preventing recurrent disease [4], and SPTs are frequently overlooked. First and foremost, we believe that other regions, except the head and neck, should not be neglected. Even if the tiniest symptom emerges in these patients, the head and neck surgeon should refer them to other experts or obtain diagnostic imaging of that specific region (usually CT). Certainly, a lot of patients have asymptomatic SPT, hence, we propose annual PET-CT or, if that is not possible, annual chest CT in all advanced laryngeal cancer patients.

Furthermore, 6% of our patients developed SPTs of other head and neck regions and, interestingly, they usually appeared later than in distant locations. We presume that radiotherapy has a dual function, or that preventing the development of recurrent disease also may contribute to the development of SPTs as a late complication [37]. We report a higher percentage of SPT in the lungs than in the head and neck region. However, pulmonary metastases were similar compared to previous publications reporting lung failures, particularly in laryngeal cancer patients, while regional head and neck failures are more common in oropharyngeal and hypopharyngeal carcinomas [38,39]. We set up a nomogram with the aim of predicting the occurrence of SPT. Unfortunately, none of the included clinical factors was linked to SPT. Nevertheless, the nomogram was very useful for overall predictions in our cohort.

In the analysis of the factors that lead to recurrences, only clinical factors are considered, and molecular prognostic factors are not included. In earlier studies, we found a tendency that patients with high IMP3 expression had a shorter time to recurrence (*p* = 0.051) and significantly worse disease-specific survival (*p* = 0.027) [40]. Additionally, patients with laryngeal squamous cell carcinoma had a considerably worse prognosis when their epidermal growth factor receptor expression was increased [41]. 

Certainly, the impact of SPTs on overall survival is minor compared to loco-regional or distant failures, but its effect should not be ignored. Our data show that most recurrence-related events occur in the first years of follow-up, while the peak incidence of SPTs is at the level of the second and third years of follow-up. Altogether, our data indicate that tumor relapse and occurrence of SPT represent independent worse prognosticators for OS and that long-term follow-ups are necessary. Subsequent studies with larger patient cohorts are warranted to better estimate which patients are at a higher risk of SPTs and may, therefore, require strict follow-ups.

## 5. Conclusions

No factors have been identified so far to represent reliable or irrefutable factors for predicting the risk of SPT development. Our data demonstrate that laryngeal cancer patients are faced with both the risk of locoregional failures as well as the development of SPTs. As both factors were independent worse prognosticators for overall survival, screening strategies need to be optimized accordingly. It is of utmost importance to identify tumor recurrence and/or SPT in a timely manner to allow for the early start of adequate treatment intending to improve life quality and survival.

## Figures and Tables

**Figure 1 jpm-13-00927-f001:**
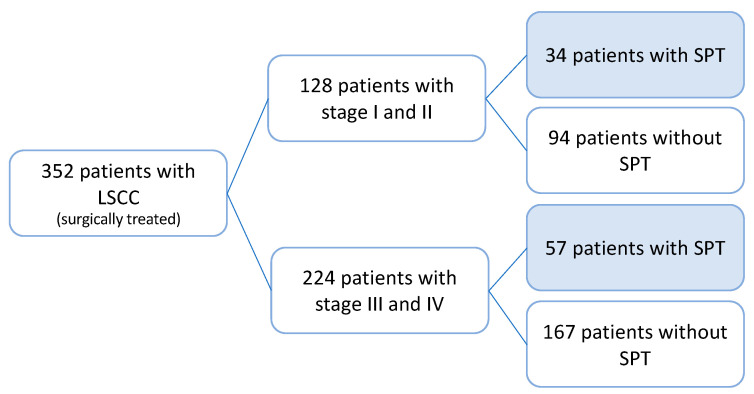
Study cohort according to tumor stage and occurrence of the secondary primary tumor. In total, 352 patients with laryngeal squamous cell carcinomas (LSCCs) were analyzed regarding the occurrence of the secondary primary tumor (SPT), including 128 patients (36.4%) with stage I–II and 224 patients (63.6%) with stage III–IV disease. Accordingly, SPT occurred in 25.4% (57/224) of stage III and IV tumors compared to 26.6% (34/128) in stage I and II cases.

**Figure 2 jpm-13-00927-f002:**
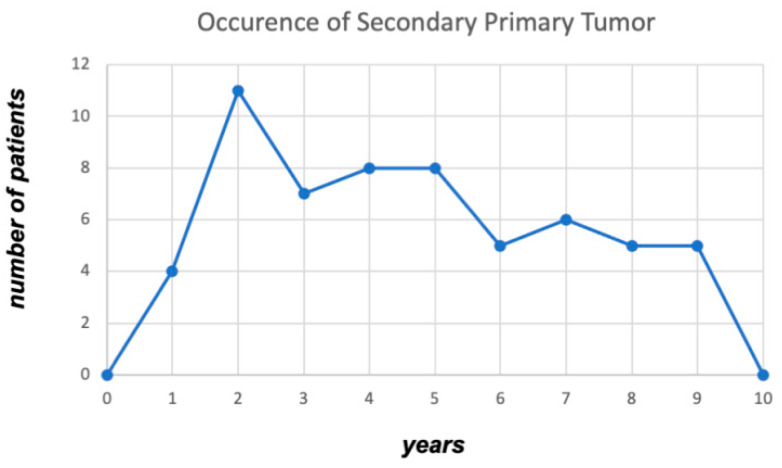
Occurrence of the secondary primary tumor. Exact data regarding the occurrence of secondary primary tumors (SPT) was available in 66 out of 91 patients (72.5%). As indicated, we observed a peak in SPT diagnosis during the second year of laryngeal cancer (*n* = 11), with treatment flattening thereafter from the third to ninth year after diagnosis.

**Figure 3 jpm-13-00927-f003:**
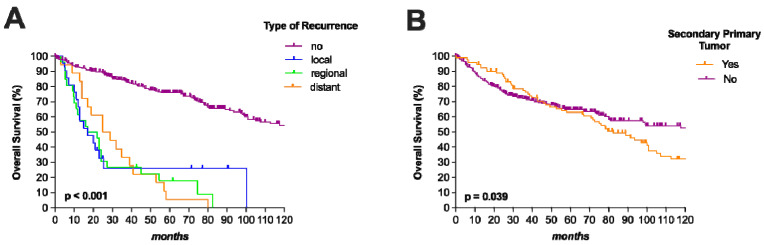
Overall survival according to the type of recurrence and secondary primary tumor. Occurrence of recurrences, regardless of which type, was associated with significantly worse overall survival (OS). The 10-year OS was around 60% in patients after treatment of laryngeal cancer and the absence of tumor relapse (**A**). Patients with secondary primary tumors showed significantly worse OS compared to those without (**B**). Interestingly, patients with absent secondary primary tumors showed similar 10-year OS rates compared to those without tumor recurrence.

**Figure 4 jpm-13-00927-f004:**
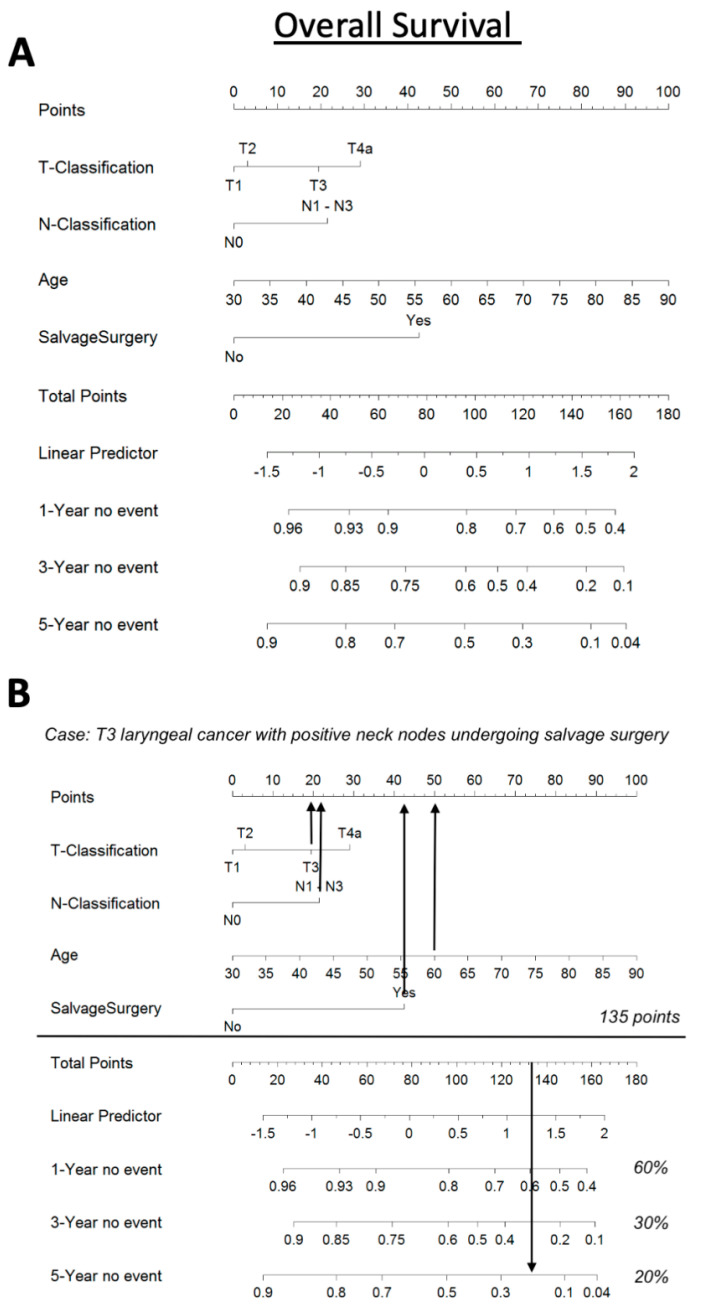
A nomogram based on age, salvage status (yes/no), T-classification (T1, T2, T3, T4a), and N-classification (N0 versus N1–3) as significant OS variables (**A**). Our nomogram predicts a 1-, 3-, and 5-year OS of 60%, 30%, and 20%, respectively, for a 60-year-old patient with T3N1 laryngeal carcinoma who recieves salvage surgery (**B**).

**Table 1 jpm-13-00927-t001:** Clinical characteristics of our patient cohort.

Characteristics	No. of Patients (%)
Sex	352 (100.0)
Female	24 (6.8)
Male	328 (93.2)
Age	
Mean ± SD	62.5 ± 9.2
Nicotin	
No	84 (23.9)
Yes	268 (76.1)
Alcohol	
No	162 (46.0)
Yes	190 (54.0)
T-classification	
T1	81 (23.0)
T2	53 (15.1)
T3	129 (36.6)
T4a	89 (25.3)
N-classification	
N0	281 (79.8)
N1	28 (8.0)
N2	40 (11.4)
N3	3 (0.9)
Tumor Stage	
Stage I	81 (23.0)
Stage II	47 (13.4)
Stage III	109 (31.0)
Stage IVa	115 (32.7)
Grading	
Gx	19 (5.4)
G1	78 (22.2)
G2	204 (58.0)
G3	51 (14.5)
Surgical margins	
R0	324 (92.0)
R1	25 (7.1)
R2	3 (0.9)
Type of Surgery	
Partial laryngectomy	102 (29.0)
Total laryngectomy	191 (54.3)
Partial laryngopharyngectomy	49 (13.9)
Total laryngopharyngectomy	10 (2.8)
Salvage Surgery	
No	308 (87.5)
Yes	44 (12.5)
PORT	
No	172 (48.9)
Yes	180 (51.1)

Abbreviations: SD, standard deviation; PORT, postoperative radiotherapy.

**Table 2 jpm-13-00927-t002:** Oncological outcome.

Outcome	No. of Patients (%)
Survival	
Alive	188 (53.4)
Dead	164 (46.6)
Cause of death	
Primary cancer	62 (17.6)
Second cancer	45 (12.8)
Other cause	57 (16.2)
Recurrence	
No	287 (81.5)
Yes	65 (18.5)
Type of recurrence	
Local	21 (6.0)
Regional	26 (7.4)
Distant	18 (5.1)
Second cancer	
No	261 (74.1)
Yes	91 (25.9)
Localization of second cancer	
Head and neck area	21 (6.0)
Lung	29 (8.2)
Other than HNC	70 (19.9)

Abbreviations: HNC, head and neck cancer.

**Table 3 jpm-13-00927-t003:** Secondary primary tumor.

	Second Head and Neck Cancer				Second Lung Cancer	
CHARACTERISTICS	No	Yes	*p*	No	Yes	*p*
	No. of patients (%)	No. of patients (%)		No. of patients (%)	No. of patients (%)	
Gender						
Male	307 (93.6)	21 (6.4)		300 (91.5)	28 (8.5)	
Female	24 (100.0)	0 (0.0)	0.201	23 (95.8)	1 (4.2)	0.707
Age						
<62.7	158 (91.3)	15 (8.7)		161 (93.1)	12 (6.9)	
≥62.7	173 (96.6)	6 (3.4)	0.043	162 (90.5)	17 (9.5)	0.441
T-classification						
T1	79 (97.5)	2 (2.5)		77 (95.1)	4 (4.9)	
T2	50 (94.3)	3 (5.7)		48 (90.6)	5 (9.4)	
T3	117 (90.3)	12 (9.3)		120 (93.0)	9 (7.0)	
T4a	85 (95.5)	4 (4.5)	0.197	78 (87.6)	11 (12.4)	0.316
N-classification						
N0	267 (95.0)	14 (5.0)		259 (92.2)	22 (7.8)	
N1	23 (82.1)	5 (17.9)		25 (89.3)	3 (10.7)	
N2	38 (95.0)	2 (5.0)		36 (90.0)	4 (10.0)	
N3	3 (100.0)	0 (0.0)	0.050	3 (100.0)	0 (0.0)	0.868
Tumor Stage						
Stage I	79 (97.5)	2 (2.5)		77 (95.1)	4 (4.9)	
Stage II	45 (95.7)	2 (4.3)		42 (89.4)	5 (10.6)	
Stage III	99 (90.8)	10 (9.2)		103 (94.5)	6 (5.5)	
Stage IVa	108 (93.9)	7 (6.1)	0.260	101 (87.8)	14 (12.2)	0.175
Smoking						
No	82 (97.6)	2 (2.4)		77 (91.7)	7 (8.3)	
Yes	249 (92.9)	19 (7.1)	0.183	246 (91.8)	22 (8.2)	1.000
Alcohol						
No	154 (95.1)	8 (4.9)				
Yes	177 (93.2)	13 (6.8)	0.505	175 (92.1)	29 (8.2)	0.847

**Table 4 jpm-13-00927-t004:** Recurrence.

CHARACTERISTICS		TYPE OF RECURRENCE			
	No	Local	Regional	Distant	*p*
	No. of patients (%)	No. of patients (%)	No. of patients (%)	No. of patients (%)	
T-classification					
T1	75 (92.6)	4 (4.9)	2 (2.5)	0 (0.0)	
T2	44 (83.0)	3 (5.7)	2 (3.8)	4 (7.5)	
T3	109 (84.5)	5 (3.9)	8 (6.2)	7 (5.4)	
T4a	59 (66.3)	9 (10.1)	14 (15.7)	7 (7.9)	0.002
N-classification					
N0	240 (85.4)	18 (6.4)	15 (5.3)	8 (2.8)	
N1	20 (71.4)	1 (3.6)	7 (25.0)	0 (0.0)	
N2	25 (62.5)	2 (5.0)	4 (10.0)	9 (22.5)	
N3	2 (66.7)	0 (0.0)	0 (0.0)	1 (33.3)	<0.001
Tumor Stage					
Stage I	75 (92.6)	4 (4.9)	2 (2.5)	0 (0.0)	
Stage II	41 (87.2)	3 (6.4)	1 (2.1)	2 (4.3)	
Stage III	95 (87.2)	4 (3.7)	7 (6.4)	3 (2.8)	
Stage IVa	76 (66.1)	10 (8.7)	16 (13.9)	13 (11.3)	<0.001
Grading					
Gx	18 (94.7)	1 (5.3)	0 (0.0)	0 (0.0)	
G1	68 (87.2)	6 (7.7)	4 (5.1)	0 (0.0)	
G2	159 (77.9)	11 (5.4)	18 (8.8)	16 (7.8	
G3	42 (82.4)	3 (5.9)	4 (7.8)	2 (3.9)	0.205
Type of Surgery					
Partial laryngectomy	97 (95.1)	3 (2.9)	2 (2.0)	0 (0.0)	
Total laryngectomy	152 (79.6)	12 (6.3)	16 (8.4)	11 (5.8)	
Partial laryngopharyngectomy	30 (61.2)	5 (10.2)	7 (14.3)	7 (14.3)	
Total laryngopharyngectomy	8 (80.0)	1 (10.0)	1 (10.0)	0 (0.0)	<0.001
Salvage Surgery					
No	261 (84.7)	11 (3.6)	21 (6.8)	15 (4.9)	
Yes	26 (59.1)	10 (22.7)	5 (11.4)	3 (6.8)	<0.001
PORT					
No	144 (83.7)	14 (8.1)	9 (5.2)	5 (2.9)	
Yes	143 (79.4)	7 (3.9)	17 (9.4)	13 (7.2)	0.043

Abbreviations: PORT, postoperative radiotherapy.

**Table 5 jpm-13-00927-t005:** Cox regression analysis.

	UNIVARIATE ANALYSIS			MULTIVARIATE ANALYSIS		
	HR	*P*	95% CI	HR	*p*	95% CI
Overall Survival						
Sex (Female)	0.643	0.225	0.32–1.31			
Age (>62.5 years)	1.536	0.009	1.11–2.12	1.610	0.007	1.14–2.27
T3–T4a versus T1–T2	2.268	<0.001	1.57–3.27	0.703	0.474	0.27–1.85
N+ versus N-	2.141	<0.001	1.52–3.01	1.299	0.205	0.87–1.95
Stage III–IV versus I–II	2.564	<0.001	1.75–3.76	3.194	0.032	1.10–9.26
G1 versus G2 versus G3	1.426	0.007	1.10–1.84	1.159	0.309	0.87–1.54
Salvage surgery (Yes)	2.874	<0.001	1.95–4.22	2.232	0.001	1.39–3.60
PORT (No)	0.691	0.024	0.50–0.95	0.831	0.383	0.55–1.26
Recurrence (Yes)	6.135	<0.001	4.31–8.70	5.050	<0.001	3.46–7.35
Second cancer (Yes)	1.397	0.040	1.02–1.92	1.441	0.039	1.02–2.04
Cancer-specific Survival						
Sex (Female)	0.924	0.878	0.34–2.55			
Age (>62.5 years)	0.883	0.630	0.54–1.46			
T3–T4a versus T1–T2	6.757	<0.001	2.91–15.6	1.792	0.431	0.42–7.69
N+ versus N−	4.237	<0.001	2.57–6.99	3.144	<0.001	1.66–5.95
Stage III–IV versus I–II	9.804	<0.001	3.56–27.0	3.144	0.215	0.51–13.2
G1 versus G2 versus G3	1.811	0.004	1.21–2.72	1.102	0.963	0.61–1.68
Salvage surgery (Yes)	5.076	<0.001	2.99–8.62	5.128	<0.001	2.66–9.80
PORT (No)	0.781	0.339	0.47–1.30			
Recurrence (Yes)	41.67	<0.001	20.4–83.3	33.33	<0.001	15.6–71.4
Second cancer (Yes)	0.362	0.007	0.17–0.76	0.518	0.098	0.27–1.12
Recurrence-free Survival						
Sex (Female)	0.886	0.815	0.32–2.44			
Age (>62.5 years)	0.886	0.626	0.54–1.44			
T3–T4a versus T1–T2	2.404	0.003	1.35–4.27	0.618	0.439	0.18–2.09
N+ versus N−	2.801	<0.001	1.67–4.65	2.227	0.006	1.26–3.92
Stage III–IV versus I–II	2.967	0.001	1.58–5.56	2.959	0.125	0.74–11.7
G1 versus G2 versus G3	1.257	0.258	0.85–1.87			
Salvage surgery (Yes)	4.115	<0.001	2.38–7.09	3.922	<0.001	2.23–6.85
PORT (No)	0.812	0.405	0.48–1.33			

Abbreviations: HR, hazard ratio; CI, confidence interval; PORT, postoperative radiotherapy.

## Data Availability

The datasets analyzed during the current study are available from the corresponding author upon reasonable request.

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
