# Peer review of "Long-Term Care and Follow-Up in Laryngeal Cancer Patients: A Multicenter Retrospective Analysis"

_jpm, 2023, doi:10.3390/jpm13060927_

Round 1
Reviewer 1 Report
Very well written manuscript.
It is noteworthy that none of the patients received chemotherapy as a part of their treatment despite advanced or recurrent disease in some. Can you please explain why? Were there no patients who refused radical surgery?
The nomograms are quite unique: did you have a chance to test the nomograms prospectively?
The proof of your efforts will be in the ability to predict the outcome by oncologic teams outside your centres.
Does the nomogram predict a second primary? Especially another H&N and lung primary?
Your spelling for nephrectomy is given as nefrectomy and gemcitabine as gemcitabin (in supplementary materials. If you feel they are acceptable, you may leave them as such.
Author Response
Dear reviewer, our answers are attached.

Reviewer 2 Report
This is a study on long-term care and follow-up in laryngeal cancer patients. It is multicenter retrospective analysis with an adequate patient sample. The study is intresting, with well organized methodlogy and clear results. It is also well written. The presentation of the nomogram is very intresting.
comments:
- lines 87-89: Moreover, a nodule localized in the middle or distal part of the lung also suggests lung metastasis, while hilar or endobronchial lesions more likely to indicate a secondary primary tumor.
Why is that? could you be more explanatory?
- lines 147-149: Most notably, secondary head and neck cancers were noticed with a mean time of 101.1 months after initial diagnosis, which was two times longer compared to a diagnosis of secondary lung cancer (47.5 months) or carcinomas of other localizations (54.8 months).
At the same time in line 174 you mention that the mean follow up was 63.8 months. How can you explain this discrepancy about the follow up period you have definied for this study?
- line 163-164: Recurrent disease was observed in 65 (18.5%) patients of which 21 (6%) patients had local, 26 (7.4%) regional, and 18 (5.1%) distant recurrence.
What is a distant recurrence?
- how can you explain the discrepancy the possitive lymph nodes to be an independent prognostic factor for cancer-specific survival and not for overal survival?
- maybe it would be beneficial if authors made in the discussion section a comment on what are the current guidelines regarding the follow up period in case of a laryngeal carcinoma so as to highlight a diffrent follow up suggestion based on their results.
- reference 22 is a self-reference that is not specifically relevant with the context of the text where this appears.
- Authors of the same team and same affiliation have a recent publication: inEur Arch Otorhinolaryngol journal (doi: 10.1007/s00405-022-07668-1) which is not included in the references and which has quite relevant objectives, methodology and conclusion. Can you please clarify whether the aformentioned study is on diffrent patients sample?
- only 30% of the references are of the last five years, while an other almost 30% of the references are older than 20 years. I would suggest that authors should be based on more current literature.
Author Response

(The authors gave the same response as above.)
